# ATRPred: A machine learning based tool for clinical decision making of anti-TNF treatment in rheumatoid arthritis patients

Bodhayan Prasad[1], Cathy McGeough[1], Amanda Eakin[1], Tan Ahmed[1], Dawn Small[2], Philip Gardiner[2], Adrian Pendleton[3], Gary Wright[3], Anthony J. Bjourson[1], David S. Gibson[1], Priyank Shukla[1]*

1 Northern Ireland Centre for Stratified Medicine (NICSM), Biomedical Sciences Research Institute, Ulster University, C-TRIC Building, Altnagelvin Area Hospital, Londonderry, United Kingdom, 2 Western Health and Social Care Trust (WHSCT), Altnagelvin Area Hospital, Londonderry, United Kingdom, 3 Belfast Health and Social Care Trust (BHSCT), Belfast City Hospital, Belfast, United Kingdom

* p.shukla@ulster.ac.uk

**Data Availability Statement:** All source code and data are available at https://github.com/ShuklaLab/ATRPred.

## Abstract

Rheumatoid arthritis (RA) is a chronic autoimmune condition, characterised by joint pain, damage and disability, which can be addressed in a high proportion of patients by timely use of targeted biologic treatments. However, the patients, non-responsive to the treatments often suffer from refractoriness of the disease, leading to poor quality of life. Additionally, the biologic treatments are expensive. We obtained plasma samples from N = 144 participants with RA, who were about to commence anti-tumour necrosis factor (anti-TNF) therapy. These samples were sent to Olink Proteomics, Uppsala, Sweden, where proximity extension assays of 4 panels, containing 92 proteins each, were performed. A total of n = 89 samples of patients passed the quality control of anti-TNF treatment response data. The preliminary analysis of plasma protein expression values suggested that the RA population could be divided into two distinct molecular sub-groups (endotypes). However, these broad groups did not predict response to anti-TNF treatment, but were significantly different in terms of gender and their disease activity. We then labelled these patients as responders (n = 60) and non-responders (n = 29) based on the change in disease activity score (DAS) after 6 months of anti-TNF treatment and applied machine learning (ML) with a rigorous 5-fold nested cross-validation scheme to filter 17 proteins that were significantly associated with the treatment response. We have developed a ML based classifier ATRPred (anti-TNF treatment response predictor), which can predict anti-TNF treatment response in RA patients with 81% accuracy, 75% sensitivity and 86% specificity. ATRPred may aid clinicians to direct anti-TNF therapy to patients most likely to receive benefit, thus save cost as well as prevent non-responsive patients from refractory consequences. ATRPred is implemented in R.

## Author summary

Rheumatoid arthritis (RA) is a chronic disease, characterised by joint pain, damage and disability. It is known to affect at least 1% of European population. It can be addressed in

**Funding:** BP acknowledges support of Vice-Chancellor's Research Scholarship (VCRS), Ulster University. AJB acknowledges support from the European Union Regional Development Fund (ERDF), EU Sustainable Competitiveness Programme for Northern Ireland & the Northern Ireland Public Health Agency (HSC R&D) and Ulster University. PS acknowledges support from the Innovate UK NxNW ICURe programme. The funders had no role in study design, data collection and analysis, decision to publish, or preparation of the manuscript.

**Competing interests:** I have read the journal's policy and the authors of this manuscript have the following competing interests: A UK-wide patent application has been filed by the Ulster University; UK Application No. 2208371.1, patent pending. All the aspects of this manuscript are covered in this patent application.

a high proportion of patients by timely use of targeted biologic treatments. But, biologic treatments continue to rank among the highest grossing drugs. Adalimumab (a biologic drug) for example, alone generated 20 billion US dollars of revenue worldwide in 2018. Additionally, European countries with limited resources, place volume controls on reimbursed medicines. A cheaper prognostic test for biologic response can help clinicians prescribe treatments to those who will receive benefit and also rationalise expensive treatments. In this study we have proposed an informative plasma protein signature, consisting of 17 proteins, and have developed a ML based classifier ATRPred (Anti-TNF Treatment Response Predictor), which can predict anti-TNF treatment response in RA patients with 81% accuracy. With this work we have tried to help clinicians to optimise treatment selection, reduce spend on biologics in unresponsive patients and overall improve quality of life for non-responsive RA patients. Our study has also identified endotypes or molecular sub-classes of RA using plasma protein profiles. These endotypes did not show difference in the responsiveness towards the anti-TNF, however they may be helpful in understanding of the disease and response to other treatments, going forward.

This is a *PLOS Computational Biology* Software paper.

## Introduction

Rheumatoid Arthritis (RA) is a chronic autoimmune condition characterised by relapsing joint pain, inflammation, and damage along with systemic effects and elevated morbidity. Without effective treatment, RA patients suffer greater risk of disability [1]. Initially, RA patients are treated with non-steroidal anti-inflammatory drugs and conventional disease modifying anti-rheumatic drugs (DMARDs). Patients, refractory to conventional DMARDs, are subsequently prescribed biologic DMARDs [2], among which anti-tumour necrosis factor (anti-TNF) therapies are common, which includes adalimumab, etanercept, infliximab, certolizumab or golimumab–a monoclonal anti-TNF antibody. However, not all patients respond well to anti-TNF therapy. Approximately 10–30% do not respond initially and 23–46% lose the responsiveness over time [3]. A recent article suggests that at least 6% of RA patients on biologics suffer from a refractory condition of the disease [4]. This suggests the existence of molecular sub-classes within the broad disease class. These molecular sub-classes are known as endotypes. Unlike, phenotype which involves only observable characteristics, an endotype has direct relation with disease process as it involves inflammatory parameters and specific biological mechanisms. A recent paper from McInnes et al [5] advocates the need for clinically meaningful RA endotypes to stratify patients for the therapeutics.

Clinicians generally decide to prescribe anti-TNF therapy based on their disease severity, progression, and other comorbidities. Recent research suggests that the clinicians often switch between different treatments empirically because of a lack of suitable predictive tests [6]. A major downside of this approach is that for patients who remain unresponsive to attempted biologic treatments, inadequate suppression of ongoing disease activity elevates the risk of permanent joint damage and disability [7]. This argues for the need of developing a better prognostic model, that can predict a patient's responsiveness towards the anti-TNF therapy.

Furthermore, RA is known to affect at least 1% of European population [8]. A recent epidemiological study has reviewed prevalence of RA in different countries of every continent and

reports that the prevalence is still close to 1% in many European countries [9]. Additionally, biologic treatments remain relatively costly and continue to rank among the highest grossing drugs. Humira (adalimumab) for example, alone generated 20 billion US dollars of revenue worldwide in 2018 [10]. A very recent study [11] has pointed out various hidden access barriers to biologic treatment in the European Union (EU).

Thus, there is a strong clinical as well as health-economic need for a more personalised prognostic models which can determine likelihood of response to anti-TNF therapy [12]. Several studies using different omics profiles have attempted to predict response to anti-TNF therapy [13]. Literature review shows that the researchers have identified serum proteomic biomarkers for response to anti-TNF therapy [14] including one based on autoantibody and cytokine profiles [15]. Biomarkers have also been found specific to infliximab drug response [16] and etanercept drug response [17]. Further differentiated responses have been noted for adalimumab and infliximab [18]. Also, clinical efficacy can be intensified with infliximab using therapeutic drug monitoring approaches [19]. Several multi-omics approaches have also been used to predict anti-TNF efficacy [20]. For example, an integrated multi-omics approach of previously known DNA, RNA, and protein biomarkers [21], and a more recent approach which combines transcriptomic and genomic analysis [22]. However, none of these studies have presented a robust scoring scheme/model for drug responsiveness that can help in decision making under a clinical setting; rather they relied on only p-values.

We have strictly followed European League Against Rheumatism (EULAR) criteria for patient recruitment, as it is known to have good construct, criterion, and discriminatory validity [23]. Further, to stratify a patient's potential response to treatment, a proteomic profile (which is highly variable) may better reflect current disease state than transcriptomic (variable) or genomic (constant) profiles. With the advent of new high-throughput proteomics technology such as multiplexed proximity extension assay (PEA), it is now possible to profile a patients' plasma proteins with high accuracy and sensitivity [24]. This study was designed to identify a robust protein signature which can predict a patient's response to anti-TNF therapy using a highly sensitive protein detection platform. This study investigates whether plausible endotypes with clinical relevance can be detected in the plasma proteome and if further stratification can predict future response to anti-TNF treatment. Machine Learning (ML) based algorithms, which have been widely exploited for prediction and/or classification problems in bioinformatics, were deployed to mine targeted proteome data. This could help clinicians to optimise treatment selection, reduce spend on biologics in unresponsive patients and overall improve quality of life for non-responsive RA patients.

## Design and implementation

### Ethics statement

Office for Research Ethics Committees Northern Ireland (ORECNI) (11/NI/0188), Ulster University Research Ethics Committee (UREC) (REC/11/0366), Belfast Health and Social Care Trust (11098AB-SS) and Western Health and Social Care Trust (WT/11/35) approvals were obtained for the study. All methods were performed in accordance with the relevant guidelines and regulations. Formal written informed consent was obtained for all participants in the study, allowing for publication of anonymised clinical data.

### Patient recruitment and selection criteria

A total of one hundred and forty-four (N = 144) Rheumatoid arthritis (RA) patients who were unresponsive to conventional DMARDs and naïve to biologic DMARDs were recruited from

rheumatology biologic clinics at Altnagelvin Hospital, Londonderry and Musgrave Park Hospital, Belfast, Northern Ireland. The study inclusion criteria were: i) RA patients fulfilling EULAR classification criteria [25,26], ii) about to receive anti-TNF treatment as part of routine clinical practice, iii) fulfil the BSR 2001 criteria for anti-TNF therapy [27], iv) had a DAS28 score of >5.1 when assessed for treatment (before baseline), and v) reached 6 months of follow-up. Patients who stopped anti-TNF temporarily during first six months or discontinued therapy prior to the 6 months' follow-up for reasons other than inefficacy were excluded.

## Sample collection and collation of clinical information

The study was supported by a patient advisory group who met regularly throughout the study to advise on study design, recruitment literature and results dissemination. Eligible patients were invited by mailed patient information sheets, a minimum of 48 hours before a routine care appointment. Written informed consent was obtained and blood samples were collected prior to anti-TNF treatment. Blood samples were then processed to plasma by centrifugation, aliquoted and stored at -80˚C until shipped to Olink Proteomics, Uppsala, Sweden for proximity extension assay (PEA) analysis. Clinical and demographic information were collated from medical records and clinic databases. The disease activity score across 28 joints (DAS-28) based on erythrocyte sedimentation rate (ESR) was recorded at baseline and after six months of anti-TNF therapy. Patients were classified as responders and non-responders at six months as per British Society for Rheumatology (BSR) response criteria [28]. Further, a patient, whose drug was changed from anti-TNF to a different class by clinicians were also classified as non-responders. Out of N = 144 patients recruited 55 were either lost to follow-up or were given other biologic DMARDs (such as Tociluzimab, Ritiuximab, etc). The recruits lost were unable to make 6 months follow-up appointments, or complete composite data required to calculate DAS score were not available.

## Plasma protein profile

Patients' plasma samples were analysed by multiplexed PEA [29] provided by Olink Proteomics (www.olink.com). Following four Proseek Multiplex panels comprising 92 proteins each were used for analyses: cardiovascular panels II and III, immune response panel and the inflammatory panel. Each panel was quantified by real-time PCR using the Fluidigm BioMark HD platform. In each panel run, 92 samples, 1 negative control and 3 positive controls were analysed. Controls were used for determining the assay limit of detection (LoD) values as well as allowing normalization of measurements into ddCq ($\Delta\Delta C_q$: double delta quantification cycle in qPCR) values. The ddCq values are then $log_2$-transformed to promote normal distribution for subsequent analysis. Olink proteomics returned protein expression data in exponential scale called normalised protein expression (NPX), such that the real expression values are proportional to $2^{NPX}$. Each protein's NPX values are relative quantification and hence they cannot be compared across different proteins [30]. Therefore, to obtain comparable results for all proteins [31] and as a pre-processing step for machine learning inputs, each of them is separately scaled into a standard normal distribution $\sim N(0, 1)$. A total of 352 proteins passed the initial quality control (QC) and were subsequently used for the statistical and machine learning based analysis.

## Statistical, computational and bioinformatics analyses

All statistical and computational analyses were carried out in R v3.6.1 [32]. The t-test or chi-square test (as appropriate) to check for statistical significance of demographic and clinical

features, and the principal component analysis (PCA) of Olink proteomics data, were performed in the *base* R package. Quality control (QC) of protein NPX datasets involved discarding protein values which were flagged with a QC warning (sample did not pass quality control for a given protein panel). Also, NPX values were removed if below the limit of detection (LoD) level for a given protein PEA, resulting in < 2% of missing values. Since missingness was very small, it was imputed using k-Nearest Neighbour (k-NN) method using the *RANN* package [33]. PCA result was validated with leave-one-out cross-validation (LOOCV) using *sinkr* package [34]. General ML pre- and post- processing methods were derived from *caret* [35] and *e1071* package [36]. Further, we deployed generalised linear models (GLMs), using the *glmnet* package [37], to create an intuitive mathematical formulation with a linear combination of protein expression values. Receiver operator characteristic (ROC) curves were obtained via *pROC* package [38]. Finally, Youden Index [39] was used to choose the best point in ROC curve to calculate thresholds for model score to obtain sensitivity and specificity values. Box plot and beeswarm plot were drawn using *beanplot* package [40] and *beeswarm* package [41] respectively, and *gplots* package [42] and *ggrepel* package [43] were used for presenting the results. The final model selection was done based on Area Under the ROC Curve (AUC) metric, which is the most preferred metric for the classification problems. Enrichment analysis and Protein-Protein Interaction (PPI) network analysis was performed using STRING database [44]. The Gene Ontology (GO) terms were summarised using REVIGO [45] with its default parameters and the PPI networks were visualised using Cytoscape [46], an open-source software commonly used for network-based analysis. The Pearson's correlation coefficient between the protein features was computed using *stats* namespace under *base* R package. This was followed by hierarchical clustering and plotting using the *heatmaply* package [47].

## Feature selection with machine learning

A total of 500 simulations were run by randomly splitting the dataset into 80%:20% and a GLM was learned on 80% training data and tested on 20% test data. If the GLM model had better than random performance (i.e., AUC > 0.5), the feature selected in the model was then appended to a feature list. Thus, the importance of a feature reflects its frequency in the feature list. For example, a frequency of 0.8 for a feature represents that the feature showed up in 80% of the 500 simulated models. It is worth mentioning here, that multiple proteomics signature, having different feature set, are possible [48]. However, getting all the signature and its performance can be computationally expensive due to large number of combinations possible. Therefore, we went with a deterministic approach of stepwise feature selection, by calculating feature importance (FI) as described above, using a fixed seed value of 200 for 500 simulations.

## Machine learning based model development

Our dataset involved 89 samples; hence we chose 5-fold double alias nested cross-validation (CV) for the development of the predictive model [49]. This CV scheme for testing ensures no bias in the selection of completely independent model-blind test-set [50]. Model evaluation was done first by having only gender and baseline DAS and then including protein features one-by-one as per the frequency obtained during feature selection in decreasing order. Mean AUC of training and test sets were measured after fitting a GLM, which was optimised for lambda hyperparameter by 10-fold CV within the training set. The GLM was an Elastic Net with alpha of 0.9, which implements regression with 90% LASSO and 10% Ridge regularization. The aim was to select non-correlated protein, which is achieved by LASSO regularization; a popular method used for feature selection. However, 10% of Ridge regularization was kept to overcome LASSO's limitation to saturate with fewer features. The protein feature set having the highest test set AUC, without the decrease in training set AUC, was selected and the model performance was noted. Finally, with

these protein features along with gender and baseline DAS, the model was trained on the whole data and the beta or regression coefficients were computed.

### ATRPred tool development

An R-based package was developed for implementing the above-mentioned ML model with the help of *devtools* package [51]. An input file template along with sample input files of a responder as well as a non-responder are also included in the examples folder present within the package. The R function *antiTNFresponse()* reads the input and normalises the same with the internal 89 patient data to get comparable numbers for feature sets and finally scores the patient for response to the anti-TNF therapy. It then calculates the patient's probability to respond anti-TNF treatment and predicts if the patient will be a responder or non-responder. This tool is provided as an open-source GitHub repository at https://github.com/ShuklaLab/ATRPred.

## Results

The main demographic and clinical features of the patients are shown in Table 1. Gender and DAS values at both baseline and 6 months, were found to be statistically significant ($p < 0.05$) between responders and non-responders. The anti-TNF response rate of 67% in our study is almost identical to the 68% reported in a larger study [52]. However, neither this study [52] nor any other study has reported any gender difference as per the author's knowledge. This deference might be due to gender selective confounders like smoking history for which unfortunately the data was not available.

### Exploratory data analysis on plasma proteins

Principal Component Analysis (PCA) for all n = 89 patients was performed to visualise potential endotypes based on plasma proteome profile. The elbow plot of first 30 PCs showed the drop of explained variance to less than 1% at PC 20 (S1A Fig). Therefore, we carried out

**Table 1. Demographic and clinical features of rheumatoid arthritis patients.** Gender and DAS values (both at baseline and 6 months) were found to be statistically significant between responders and non-responders. RF = Rheumatoid Factor, ACPA = Anti-citrullinated protein/peptide antibody, Anti-CCP = Anti-cyclic citrullinated peptides, DMARD = Disease-modifying antirheumatic drugs and DAS28-ESR = Disease activity score with 28-joint counts and erythrocyte sedimentation rate.

| Cohort Characteristics | Responders (N = 60) | Non-Responders (N = 29) | Combined (N = 89) | P-value |
|---|---|---|---|---|
| Gender, female, n (%) | 51 (85.0) | 17 (58.6) | 68 (76.4) | *0.006 |
| Age at baseline, mean (s.d.), years | 60.6 (11.8) | 61.1 (10.3) | 60.8 (11.3) | 0.848 |
| Disease duration, mean (s.d.), years | 8.7 (7.9) | 11.1 (10.8) | 9.5 (9.0) | 0.299 |
| RF Seropositivity, n (N)# | 38 (48) | 18 (25) | 56 (73) | 0.65 |
| ACPA/anti-CCP Seropositivity, n (N)# | 34 (42) | 16 (23) | 50 (65) | 0.46 |
| Concurrent conventional DMARD at baseline, n (%) | 55 (91.6) | 26 (89.7) | 81 (91.0) | - |
| Concurrent conventional DMARD at 6 months, n (%) | 38 (63.3) | 14 (48.3) | 52 (58.4) | - |
| Adalimumab, n (%) | 40 (66.7) | 12 (41.4) | 52 (58.4) | - |
| Etanercept, n (%) | 17 (28.3) | 12 (41.4) | 29 (32.6) | - |
| Infliximab, n (%) | 0 (0.0) | 1 (3.4) | 1 (1.1) | - |
| Certolizumab, n (%) | 2 (3.3) | 2 (6.9) | 4 (4.5) | - |
| Golimumab, n (%) | 1 (1.7) | 2 (6.9) | 3 (3.4) | - |
| DAS28-ESR at baseline, mean (s.d.) | 5.7 (1.2) | 4.8 (1.4) | 5.4 (1.3) | *0.006 |
| ΔDAS28-ESR at 6 months, mean (s.d.) | -3.0 (1.1) | -0.2 (1.1) | -2.1 (1.7) | *4.8e-14 |

*significant ($p < 0.05$)

#where data was available

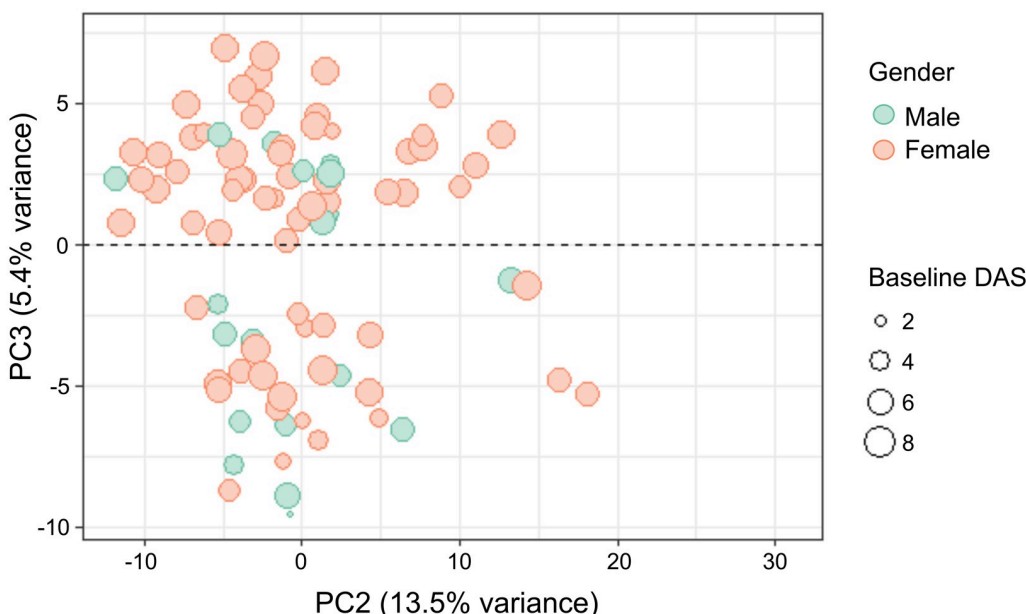

**Fig 1. Principal component analysis (PCA) plot of rheumatoid arthritis patients (n = 89) using 352 plasma protein Normalised Protein Expression (NPX) values reveals two molecular sub-classes or endotypes with respect to positive and negative third principal component (PC3) values.** Endotype 1 is with PC3 values > 0 and endotype 2 is with PC3 values < 0. Each data point represents a patient, where size of the dot is proportional to the disease activity score (DAS) of the patient at baseline.

LOOCV of the first 20 PCs, which gave top 20, 6, and 4 PCs with minimum predicted sum of squares (PRESS) for naïve, approximate, and pseudoinverse approaches, respectively (S1B Fig). Although the naïve approach has limitations [53], all three LOOCV approaches suggested that at least first 4 PCs are important. The first two principal components (PC1 and PC2) did not show any segregation; however, the third principal component (PC3) was able to subdivide patients into two distinct clusters i.e., endotypes (Fig 1). The demographic and clinical features for each cluster are shown in Table 2. A statistically significant difference (p < 0.05) in baseline DAS and gender was noted between the two clusters. Age, disease duration and anti-TNF biologic treatment response were not significantly different between the two clusters. The association between baseline DAS and gender within the clusters is illustrated in Fig 1. The plot indicates a relatively higher baseline DAS and a higher proportion of females in the cluster positioned in the upper/positive PC3 quadrant. It appears that the two endotypes clearly distinguish patients based on disease activity and are gender dependent.

**Table 2. Demographic and clinical features of two molecular sub-class or endotypes presented in Fig 1.** Gender and baseline DAS values were found to be statistically significant between the two endotypes. DAS28-ESR = Disease activity score with 28-joint counts and erythrocyte sedimentation rate.

| Cohort Characteristics | Endotype 1 (N = 55) | Endotype 2 (N = 34) | P-value |
|---|---|---|---|
| Gender, female, n (%) | 46 (83.6) | 22 (64.7) | *0.041 |
| Age at baseline, mean (s.d.), years | 61.2 (11.1) | 60.1 (11.8) | 0.648 |
| Disease duration, mean (s.d.), years | 10.1 (8.5) | 8.5 (9.8) | 0.480 |
| DAS28-ESR at baseline, mean (s.d.) | 5.7 (1.1) | 5.0 (1.4) | *0.022 |
| ΔDAS28-ESR at 6 months, mean (s.d.) | -2.3 (1.6) | -1.9 (1.8) | 0.248 |
| Responders, n (%) | 38 (64.7) | 22 (69.1) | 0.668 |

*significant (p < 0.05)

## Anti-TNF response feature selection and classifier

A quick summary of the computational pipeline built for the discovery of plasma protein signature is presented in Fig 2A and the detailed ML analysis schema for model development is presented in Fig 2B; both are discussed in more detail in methods section. The feature set available for building the ML classifier includes demographic and clinical data (*viz.* gender, age, disease duration, baseline DAS (BLDAS) and ΔDAS at 6 months) as well as 352 QC passed proteins' normalised NPX values. Since gender and BLDAS were found to be statistically significant to response to anti-TNF therapy as per Table 1, these two features were also included in the signature formulation.

The Feature Importance (FI) of top 30 proteins, along with gender and BLDAS is shown in Fig 3A. The graph depicting mean AUC for training as well as test set for each stepwise addition of protein features up to 30 proteins is shown in Fig 3B. The threshold of 30 proteins as features was decided after noting the gradual dip in the AUC values for test set (Fig 3B). A set of 17 protein gave the maximum mean AUC of 0.86 on test sets, without decreasing the training set AUC. The ROC curves for 5-fold training sets and test sets are shown in Fig 3C and 3D, respectively. The corresponding best point threshold on ROC curve gave a mean sensitivity of 0.75 and mean specificity of 0.86 on the test sets. The overall mean accuracy was 0.81 on test set. Further, the mean Matthews correlation coefficient (MCC), popularly used and advocated to assess the quality of binary classification [54], was 0.60, implying a good prediction for each class, *viz.* responders and non-responders. The summary of mean performance metrics is presented in S1 Table. The final model was trained on the whole dataset and mathematical formulation is presented in the next section.

## Plasma protein model for clinical decision making

The final model was trained on whole dataset and the beta coefficient of each feature obtained from the model was plotted against its feature importance (FI) obtained from the feature selection procedure and presented as Fig 4A. Table 3 summarises all the model features; gender, BLDAS and seventeen selected proteins along with their Uniprot and Entrez gene IDs, gene names, Feature Importance (FI) and Effect Sizes (ES) or regression/beta coefficients. Further the boxplot of calculated scores along with p-value for the patients is shown in Fig 4B. The model score (*S*) for each patient is given by:

$$S = \sum_{i=1}^{n} \beta_i x_i + b$$

Where, $x_i$ are model features, $\beta_i$ are corresponding effect sizes (or regression/beta coefficients) and $b$ is the intercept (or bias). Finally, the patient's response to anti-TNF can be binarised, i.e., 0 for NR and 1 for R, by choosing a threshold (*t*) and mapping the score to logistic function, which takes the output to a probability of response by patient, $p \in [0,1]$ as per:

$$p = logit(S - t) = \frac{1}{1 + e^{-(S-t)}}$$

Where *t* is the best point threshold, which was found to be 0.7136 (Fig 4B).

## Enrichment analysis with Gene Ontology (GO) terms and KEGG pathways

The 17 protein set, when tested for enrichment with Gene Ontology (GO) terms for Biological Process (BP) using STRING database, gave 72 significant (FDR < 0.05) hits as shown in S2 Table. These 72 GO BP terms along with its FDR, when summarised using REVIGO (S3 Table), were mostly involved with inflammatory response or its regulation (S2 Fig). The

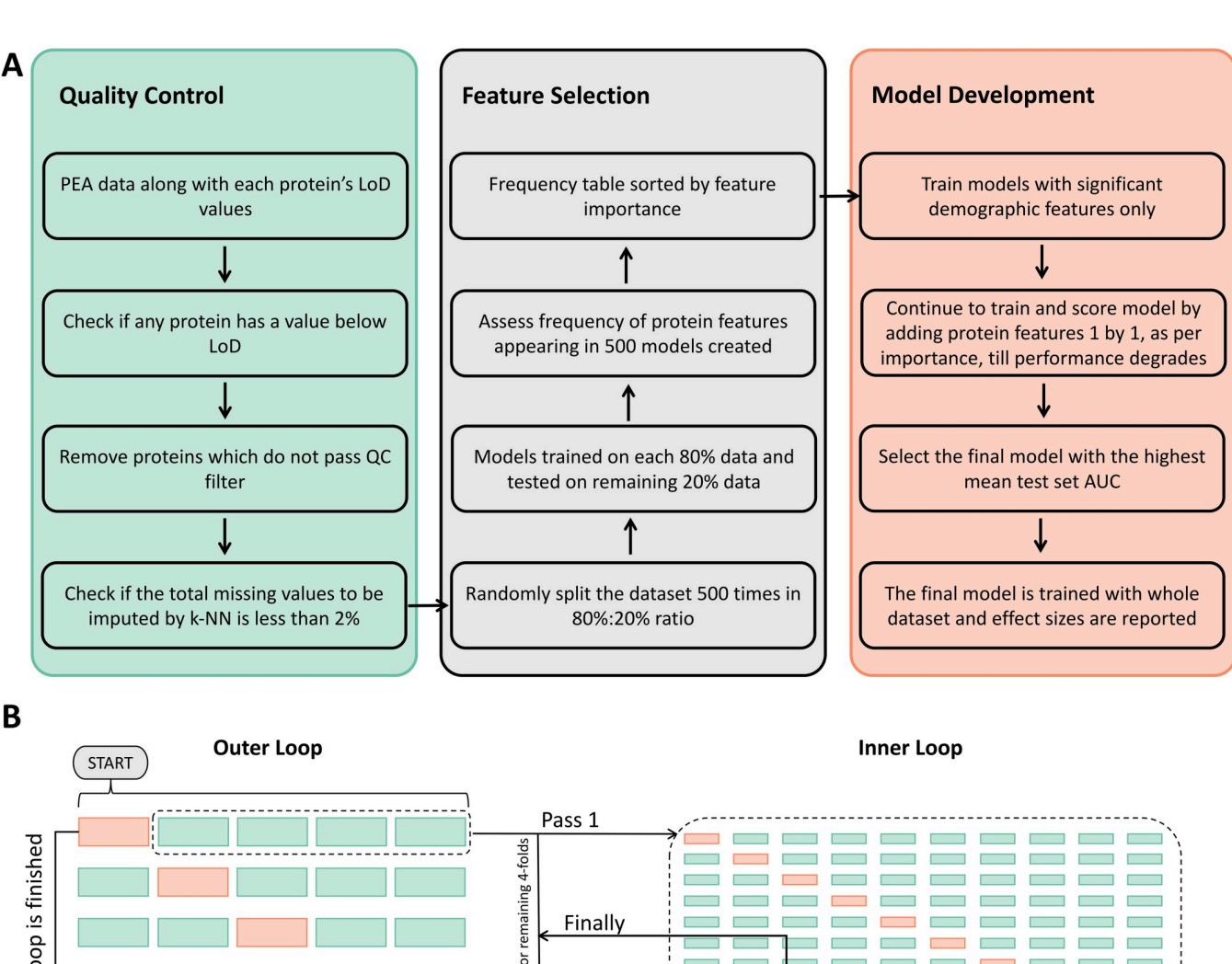

**Fig 2. (A)** Computational pipeline for the development of plasma protein signature. PEA = Protein Expression Analysis, LoD = Limit of Detection, QC = Quality Control, k-NN = k Nearest Neighbour, AUC = Area Under the Curve. **(B)** The Machine Learning (ML) schema. 5-fold nested cross-validation (CV) followed for building the classifier for response to anti-tumour necrosis factor (anti-TNF) treatment in rheumatoid arthritis (RA) patients.

enrichment for GO terms for Molecular Function (MF) gave 8 significant (FDR < 0.05) hits (S4 Table), mostly corresponding to receptor binding. Furthermore, the enrichment for GO terms for Cellular Components (CC) gave 4 significant (FDR < 0.05) hits (S5 Table), mostly suggesting extracellular region as the location of proteins. Finally, the enrichment analysis for

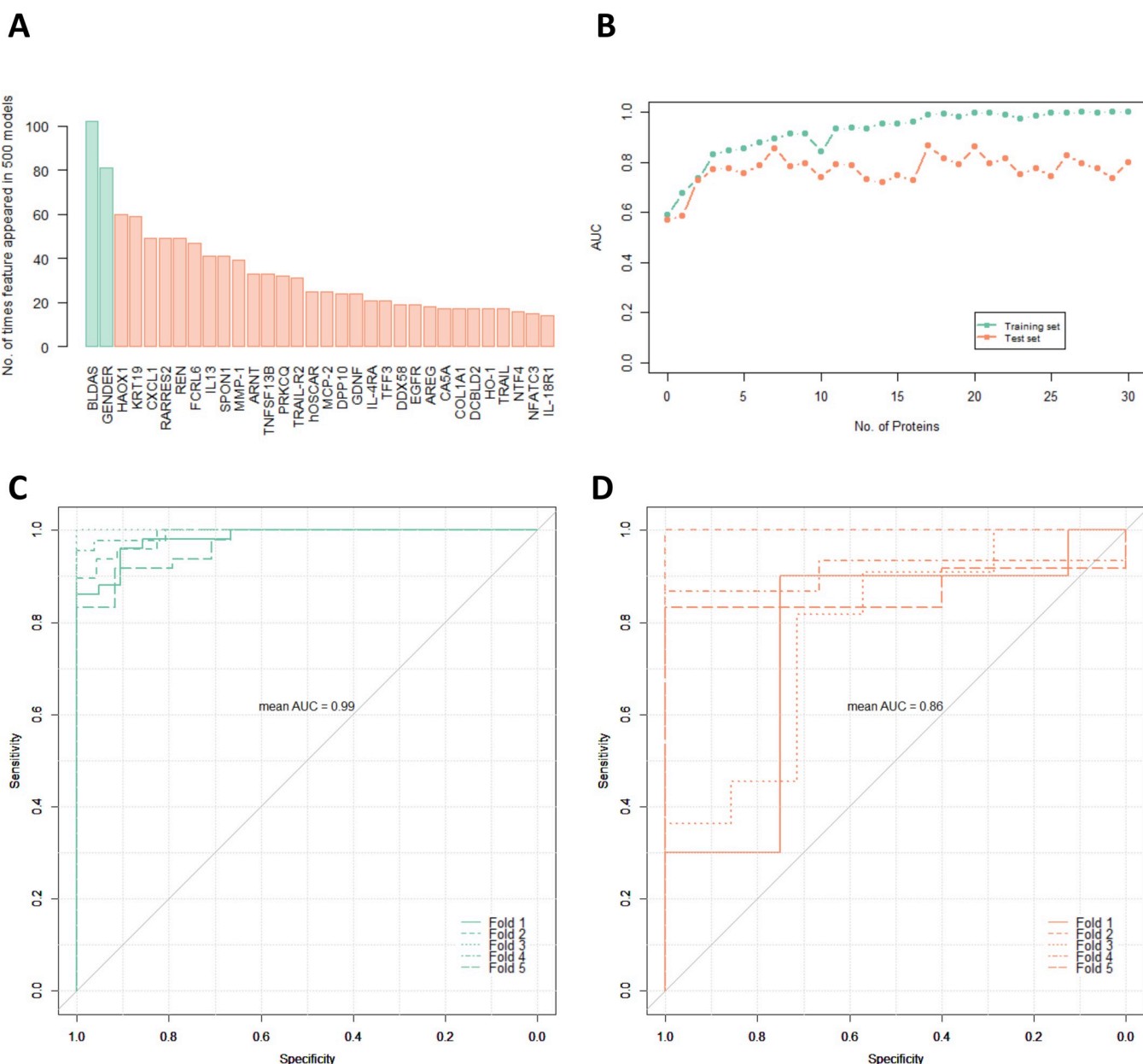

**Fig 3.** **(A)** Feature importance of top 30 proteins along with significant demographic and clinical features, viz. gender and base line disease activity score (BLDAS). **(B)** Area Under the Curve (AUC) of training and test set vs. number of protein features. A set of 17 proteins along with gender and BLDAS gave the maximum mean AUC of 0.86 on test set without decreasing the training set's AUC. Receiver Operator Characteristics (ROC) for the 5-fold cross-validation using gender, BLDAS, and 17 protein features of **(C)** training sets and corresponding **(D)** test sets.

the KEGG pathway gave 6 significant (FDR < 0.05) hits as shown in S6 Table. These hits include, as expected, rheumatoid arthritis pathway. Further, it also included IL-17 signalling pathway as well as NF-kappa B signalling pathway, which are well known for their role in inflammatory response in case of rheumatoid arthritis [55,56], suggesting their pathological role in response to biologic DMARDs as well. It was also interesting to see Measles appearing in these hits. It was recently found through pathway and network analyses of Genome-Wide Association Studies (GWAS) that Measles truly contributes to rheumatoid arthritis [57].

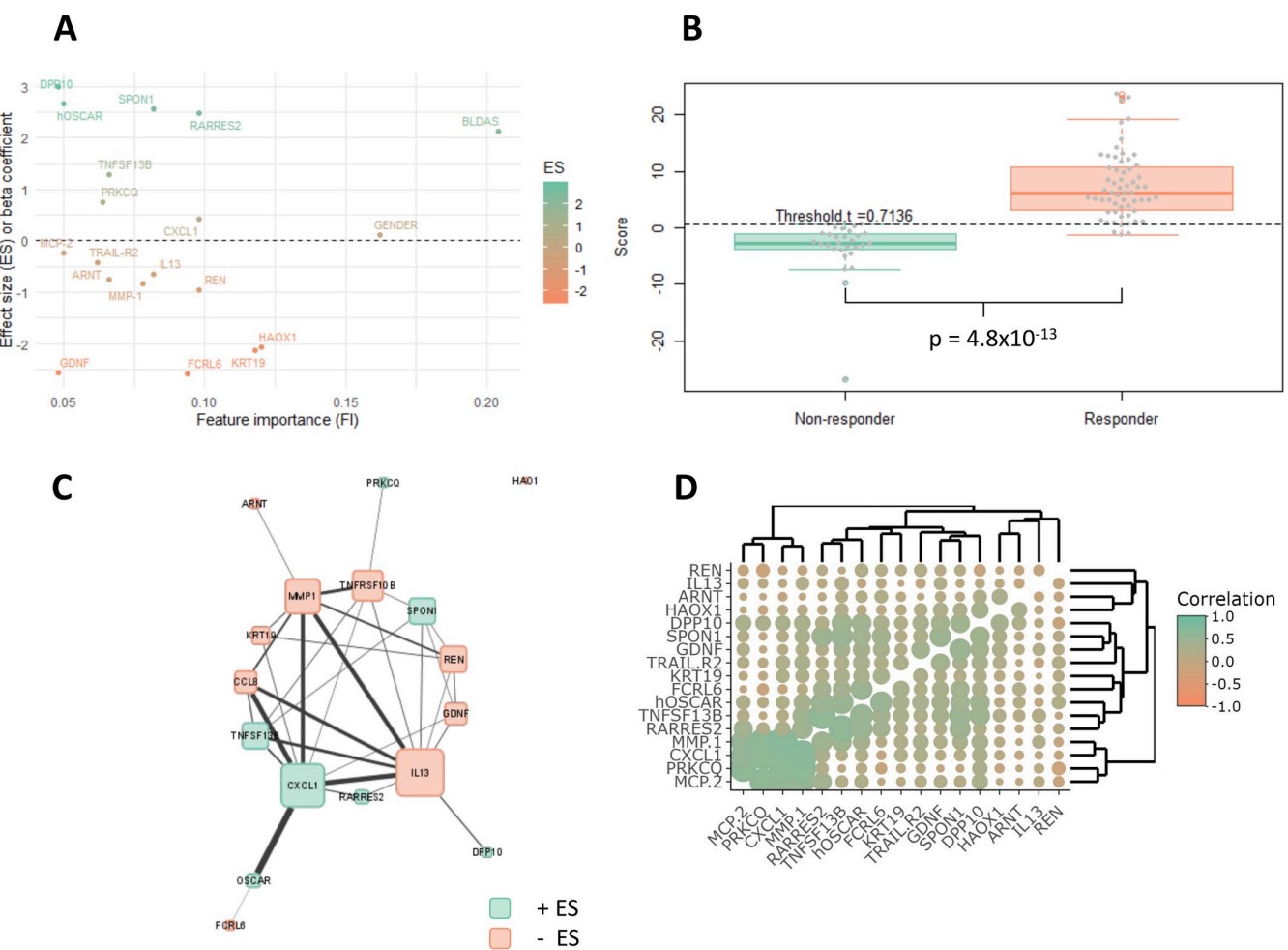

**Fig 4. (A)** Effect sizes (ES) or beta coefficients of regression vs. feature importance, i.e. fraction of 500 models, the feature appeared. **(B)** Boxplot of model score of each patient. NR = Non-responder, R = Responder. **(C)** Protein-Protein Interaction (PPI) network obtained from STRING database for 17 featured proteins. The size of the cell depicts the degree of the node i.e. number of connection with the other proteins, whereas the edge thickness represents the STRING database's interaction scores. ES = effect size, as presented in Table 3. **(D)** Pearson's correlation coefficient plot of 17 feature proteins. The size of circle depicts the -log$_{10}$(p-value) of the correlation.

## Network analysis

STRING database reports scores for Protein-Protein Interaction (PPI). These scores range from 0 for no evidence of interaction to 1 implying evidence of strong interaction. These scores are computed using different parameters such as co-expression, annotated pathways, neighbourhood, text mining, etc. We obtained the combined PPI scores of all combination of our feature proteins. The PPI network thus obtained, was then uploaded in Cystoscope for visualizing the graph in circular layout (Fig 4C). The size of the cell corresponds to the degree i.e., number of connections with the other proteins. We note that the cytokine IL13 has the highest degree of connection in the network; connected to 10 other feature proteins (Fig 4C). This was closely followed by CXCL1 which was connected to 9 other feature proteins. Further, the edge thickness is proportional to the score from STRING database. Fig 4C shows thick edges connecting IL13, CXCL1, CCL8 (alias MCP-2) and MMP1, thus implying high interaction between them. Interestingly, all these proteins are present in the extracellular region (S5 Table) and except CCL8 all other proteins are involved in IL17 signalling pathway (S6 Table).

**Table 3. Plasma protein signature, along with gender and baseline DAS (BLDAS) for anti-TNF treatment response prediction.** Feature Importance (FI) is defined as the fraction of models a feature appears in. Beta (β) Coefficients are the effect sizes of features obtained from the logistic regression analysis. DAS = Disease activity score.

| Features | Uniprot ID | Entrez Gene ID | Gene Name | Olink Panel | FI | β Coeff. |
|---|---|---|---|---|---|---|
| Intercept or bias, b | - | - | - | - | - | 3.800 |
| Baseline DAS (BLDAS) | - | - | - | - | 0.21 | 2.133 |
| Gender (M:1; F:0) | - | - | - | - | 0.17 | 0.116 |
| KRT19 | P08727 | 3880 | Keratin 19 | IMMUNE | 0.13 | -2.126 |
| HAOX1/HAO1 | Q9UJM8 | 54363 | Hydroxyacid oxidase 1 | CVD II | 0.13 | -2.068 |
| CXCL1 | P09341 | 2919 | C-X-C motif chemokine ligand 1 | CVD II + INFLAM | 0.10 | 0.421 |
| RARRES2 | Q99969 | 5919 | Retinoic acid receptor responder 2 | CVD III | 0.10 | 2.488 |
| FCRL6 | Q6DN72 | 343413 | Fc receptor like 6 | IMMUNE | 0.10 | -2.595 |
| REN | P00797 | 5972 | Renin | CVD II | 0.10 | -0.960 |
| IL13 | P35225 | 3596 | Interleukin 13 | INFLAM | 0.09 | -0.651 |
| SPON1 | Q9HCB6 | 10418 | Spondin 1 | CVD III | 0.08 | 2.557 |
| MMP-1/MMP1 | P03956 | 4312 | Matrix metallopeptidase 1 | INFLAM | 0.08 | -0.830 |
| ARNT | P27540 | 405 | Aryl hydrocarbon receptor nuclear translocator | IMMUNE | 0.07 | -0.758 |
| TNFSF13B | Q9Y275 | 10673 | Tumor necrosis factor superfamily member 13b | CVD III | 0.07 | 1.281 |
| PRKCQ | Q04759 | 5588 | Protein kinase C theta | IMMUNE | 0.07 | 0.744 |
| TRAIL-R2/TNFRSF10B | O14763 | 8795 | TNF receptor superfamily member 10b | CVD II | 0.07 | -0.421 |
| hOSCAR/OSCAR | Q8IYS5 | 126014 | Osteoclast associated, immunoglobulin-like receptor | CVD II | 0.05 | 2.661 |
| MCP-2/CCL8 | P80075 | 6355 | C-C motif chemokine ligand 8 | INFLAM | 0.05 | -0.243 |
| DPP10 | Q8N608 | 57628 | Dipeptidyl peptidase like 10 | IMMUNE | 0.05 | 2.990 |
| GDNF | P39905 | 2668 | Glial cell derived neurotrophic factor | INFLAM | 0.05 | -2.574 |

Out of these four highly interactive proteins, only CXCL1 has positive effect size to response to treatment, whereas IL13, CCL8, and MMP1 have negative effect sizes (Table 3). Thus, a high expression of CXCL1 and low expression of IL13, CCL8, and MMP1 will lead to a better response to anti-TNF treatment. Further, these four highly interacting proteins have smaller effect sizes compared to other proteins (Fig 4A), suggesting they are correlated due to their high PPI scores. We confirmed that indeed MMP1, MCP-2 (alias CCL8) and CXCL1 are significantly and highly correlated (Fig 4D). The elastic net regression distributes the weightage among the three proteins due to redundancy, as these variables have similar variations. On the contrary, less correlated features, even if they have low FI, have high effect sizes, since they have independent variation and can contribute more to anti-TNF treatment response prediction.

## Discussion

Rheumatoid arthritis (RA) patients show different pathologies in terms of functional or biological mechanism, treatment response, etc. and hence can be considered as a broad disease class containing different disease entity or sub-class. Therefore, there is a need to further stratify patients based on their distinct functional or pathobiological mechanism, more commonly called as endotypes [58]. A recent review article [59], investigates such pathobiological endotypes in early RA (n = 85). They validated 2 proteins, 52 SNPs and 72 gene expression biomarkers, that were predictive of changes in DAS28-CRP, identified from literature review. Out of the 72 transcript biomarkers, they independently replicated 8 biomarkers (SORBS3, AKAP9, CYP4F12, MUSTN, CX3CR1, SLC2A3, C21orf58 and TBC1D8). Further, the two protein candidates viz. sICAM1 and CXCL13 were also validated as predictor of anti-TNF response. They have also validated 2 SNPs (rs6028945 and rs73055646), that were significantly

associated with anti-TNF response. Using 11 biomarkers, this integrative approach showed an anti-TNF response predictability with an AUC of 0.815.

The current study uncovered two distinct endotypes based on the expression profile of 352 plasma proteins, which had significantly different gender proportions and baseline DAS (Fig 1 and Table 2). Since these endotypes were not significantly different in terms of their anti-TNF treatment response (Table 2), there is a possibility of the existence of two distinct RA disease endotypes, which may be important in other aspects of the disease management or other drug response.

Gender is known to be significantly associated with plasma protein profile [60]. Further, DAS28 is also known to be correlated with the plasma proteins such as IL37 [61] and CXCL10 [62]. A significantly higher average ESR has been observed in females of age up to 75 years [63]. Considering the above literature, there is another possibility that the two endotypes uncovered in this study may be totally unrelated to RA. Hence, the clinicians may consider keeping a strict vigil on these endotypes, which may be helpful in better informed decision making.

Anti-TNF therapy is also a part of treatment regimens followed in other inflammatory disorders like psoriatic arthritis and inflammatory bowel disease (IBD), which includes Crohn's disease (CD) and ulcerative colitis (UC). Proteomic signature for response to anti-TNF treatment in these disorders have also been studied. About 57 out of 107 targeted proteins were found to be predictive to anti-TNF treatment response with AUC of 0.76 in psoriatic arthritis [64]. In another study [65], 25 potential anti-TNF treatment predictive biomarkers based on significant differential expression between good and poor response were suggested out of 119 investigated proteins in psoriatic arthritis (n = 12). They further went on to investigate 4 out of the 25 proteins as the anti-TNF treatment predictive biomarkers, however, none of these 25 differentially expressed proteins have any intersection with our feature proteins. Another study [66] tried to stratify patients (n = 56) for prognosis or predicting response to anti-TNF therapy in IBD by identifying candidate proteomics biomarkers involved in therapeutic pathways. They suggested overall expression of defensin-5α and eosinophil cationic protein was related to responders (n = 25) and high expression of cathepsin, IL-12, IL17A and TNF was related to non-response (n = 31). Unfortunately, performance of anti-TNF treatment response prediction was not reported. With AUC of 0.86 for a relatively bigger cohort (n = 89), our plasma protein signature for the prognosis of anti-TNF therapy responsiveness in RA patients is different and its prediction performance is more accurate than of those described in the studies discussed above.

A robust machine learning based bioinformatics study requires a complete independent test set from the cross-validation set for the evaluation of the predictive model. Conventionally, single choice of independent test set is implemented, leading to possible biasness towards better performance of the predictive model. To mitigate this issue and being conscious of our limited sample size, we implemented a double or nested cross-validation based ML architecture (Fig 2B), which not only ensures an independent test set from the cross-validation sets, but also removes the biasness from choosing the independent test set by averaging the performance for all possible choice of independent test sets.

The feature importance (FI) for the proteins, obtained from the feature selection procedure, suggest the need for the feature to be included in the model. Further, the effect sizes or regression/beta coefficient, obtained from the model training, suggests the contribution of a particular feature protein has on the final score of the patient. However, FI and β-coefficient are not correlated (Fig 4A). This is due to the fact that some of the proteins are interacting with each other (Fig 4C) and therefore are correlated (Fig 4D). All the feature proteins having a lower β-coefficients are mostly correlated with each other and therefore the Elastic-Net regression

analysis distributes their weightage due to redundancy. Proteins that can classify patients into responders and non-responders to anti-TNF drugs were filtered down to seventeen (Table 3). The model presented is a simple linear combination of gender, BLDAS, and plasma protein expression values that has been implemented to develop a R-based tool ATRPred. Further, the model was 5-fold cross-validated and the mean performance was reported, which although modest, is the highest till date as per the literature review presented and the author's knowledge.

In current clinical practice, RA patients who may not respond to conventional DMARDs are routinely administered anti-TNF therapy, without enough prior knowledge of potential for efficacy. Table 3 indicates that gender and BLDAS have the highest discriminatory feature importance with respect to future response to anti-TNF therapy. These two features were also significantly different for treatment response to anti-TNF therapy (Table 1). It is common knowledge amongst clinicians that the response to biologics is greater when the ESR is higher. This knowledge is also advocated by NICE (National Institute for Health and Care Excellence) guidelines which recommends a cut-off of DAS28-ESR >5.1. The patients had all fulfilled the criteria (DAS28 >5.1) but at the time they started therapy their disease could have been going through a flare or a dip in disease activity. The former would clearly be expected to respond better, partly from the 'regression to the mean' trend. However, significance of female patients in general respond better to biologics than male patients has not been widely reported. Females are less likely to achieve remission with DAS28-ESR partly due to differences in the baseline ESR and the way the DAS28 is calculated [52]. Further, it is known that RA is more commonly found in women than men [67]. In line with this, most of the patients observed by the clinicians in our BioRA cohort were also females (Table 1). We have taken these two demographic and clinical features, viz. gender and BLDAS, as confounders and included in our signature summarised in Table 3. As per the model performance (S1 Table), we can note that the performance using just the gender and BLDAS has a test set 5-fold mean AUC of 0.57. A random model has an AUC 0.5, hence the clinical decision making using these two demographic and clinical features is only slightly better than random. However, inclusion of the 17 informative plasma proteins increased the test set 5-fold mean AUC to 0.86, resulting in about 51% increase in performance (S1 Table). Thus, our plasma protein signature may prove to be an advancement in the current clinical decision making and treatment regime of anti-TNF therapy for RA patients.

Different genome wide association studies clearly implicate the central role of the immune system in RA. To further investigate the pathways defining the patients' responsiveness and to understand the biological processes underlying the 17 protein signature, we went on to carry out enrichment analysis and network analysis. Well known rheumatoid arthritis related pathways such as IL-17 and NF-kappa B signalling pathway were found to be significantly enriched in this protein signature. Further, the clustering of significant GO BP terms for the 17 featured protein set suggests that they mostly belong to either inflammatory response or its regulation (S2 Fig). However, our study was limited to the set of proteins obtained from four pre-selected Olink Proteomics' panels; so, there is a possibility of selection bias which would influence enrichment analysis. To get an unbiased pathway topology, we extracted a protein-protein interaction network that was built on pre-existing knowledge (Fig 4C). We identified four highly interacting proteins IL13, CXCL1, CCL8, and MMP1. IL13, CXCL1 and MMP1 are involved in IL-17 signalling pathway, and their signature in responders suggests a potential role of IL-17 signalling pathway in anti-TNF response. Out of these proteins, only CXCL1 has positive effect size i.e., its higher baseline expression is indicative of future anti-TNF response. Further, CXCL1 is known to contribute to inflammation and present at higher levels during

inflammatory flare [68]. Thus, a high pre-treatment CXCL1 expression may act as a sentinel of future good response towards anti-TNF treatment.

We have identified two clusters (Fig 1 and Table 2) driven by plasma protein profile as a plausible endotypes. Unfortunately, they do not correspond to anti-TNF therapy responsiveness, but they are still significantly different in terms of disease activity and gender, and thus possibly play an important role in patient management. For example, since these endotypes are independent of future treatment response, they may indicate pre-biologic treatment pathology sub-groups, which can be investigated in future studies. Further, we have built a ML based classifier ATRPred to predict anti-TNF treatment response of RA patients at earlier timepoint using seventeen proteins feature set along with gender and BLDAS. Our model was rigorously cross-validated and performance on model-blind test sets have been presented. We have provided this tool in the form of a R-based package on an open-source GitHub repository at https://github.com/ShuklaLab/ATRPred, which may aid clinicians in deciding about putting an RA patient under anti-TNF therapy. This will help in saving the treatment cost as well as preventing nonresponsive patients to go through refractory condition of the disease leading to poor quality of life.

## Availability and future directions

ATRPred tool is built in R and provided as an open-source GitHub repository at https://github.com/ShuklaLab/ATRPred. A README file has been provided with the instructions for how to install the package and run the tool. All the R scripts and raw data used in the analysis and development of ATRPred have been included in the scripts and raw data folders present within the package. An input file template along with sample input files of a responder as well as a non-responder are also included in the examples folder present within the package. The R function *antiTNFresponse()* reads the input and normalises the same with the internal 89 patient data to get comparable numbers for feature set and finally scores the patient for response to the anti-TNF therapy. It then calculates the patient's probability to respond anti-TNF treatment and predicts if the patient will be a responder or non-responder. ATRPred may aid clinicians to optimise treatment selection, reduce spend on biologics in unresponsive patients and overall improve quality of life for non-responsive RA patients.

## Supporting information

**S1 Fig. (A)** Elbow plot for first 30 Principal Components (PCs). Dotted line represents the cut-off of 1% explained variance, crossing between PC 19 and 20. **(B)** Predicted sum of squares (PRESS) vs. number of PCs for first 20 PCs. Solid dot represents minimum value of PRESS.
(TIF)

**S2 Fig. TreeMap summary view of significant Gene Ontology (GO) Biological Process (BP) terms for the 17 featured protein set.** Size of each rectangle represents $\log_{10}$ p-value of the GO terms.
(TIF)

**S1 Table. The ML classifier performance with 5-fold nested cross-validation and the inclusion of protein features one-by-one with decreasing feature importance along with baseline DAS and gender information.** The best model performance with 17 protein features along with baseline DAS and gender information is highlighted in grey.
(DOCX)

**S2 Table. Enrichment analysis of Gene Ontology terms (Biological Process).**
(DOCX)

**S3 Table. REVIGO summary analysis of Gene Ontology terms (Biological Process).** (DOCX)

**S4 Table. Enrichment analysis of Gene Ontology terms (Molecular Function).** (DOCX)

**S5 Table. Enrichment analysis of Gene Ontology terms (Cellular Component).** (DOCX)

**S6 Table. Enrichment analysis of KEGG Pathways.** (DOCX)

## Author Contributions

**Conceptualization:** Cathy McGeough, Gary Wright, Anthony J. Bjourson, David S. Gibson, Priyank Shukla.

**Data curation:** Amanda Eakin, Tan Ahmed.

**Formal analysis:** Bodhayan Prasad, Amanda Eakin, David S. Gibson.

**Funding acquisition:** Anthony J. Bjourson, Priyank Shukla.

**Investigation:** Cathy McGeough, Amanda Eakin, Tan Ahmed, Dawn Small, Philip Gardiner, Adrian Pendleton, Gary Wright, David S. Gibson.

**Methodology:** Bodhayan Prasad, Anthony J. Bjourson, David S. Gibson, Priyank Shukla.

**Project administration:** Anthony J. Bjourson, David S. Gibson, Priyank Shukla.

**Resources:** Dawn Small, Philip Gardiner, Adrian Pendleton, Gary Wright.

**Software:** Bodhayan Prasad, Priyank Shukla.

**Supervision:** Anthony J. Bjourson, David S. Gibson, Priyank Shukla.

**Visualization:** Bodhayan Prasad, Priyank Shukla.

**Writing – original draft:** Bodhayan Prasad.

**Writing – review & editing:** Philip Gardiner, Adrian Pendleton, Gary Wright, Anthony J. Bjourson, David S. Gibson, Priyank Shukla.

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
