## [Decision Letter · Decision Letter 0]

4 Feb 2022

Dear Dr. Shukla,

Thank you very much for submitting your manuscript "ATRPred: a machine learning based tool for clinical decision making of anti-TNF treatment in rheumatoid arthritis patients" for consideration at PLOS Computational Biology.

As with all papers reviewed by the journal, your manuscript was reviewed by members of the editorial board and by several independent reviewers. In light of the reviews (below this email), we would like to invite the resubmission of a significantly-revised version that takes into account the reviewers' comments.

We cannot make any decision about publication until we have seen the revised manuscript and your response to the reviewers' comments. Your revised manuscript is also likely to be sent to reviewers for further evaluation.

Sincerely,

Manja Marz

Software Editor

PLOS Computational Biology

Manja Marz

Software Editor

PLOS Computational Biology

Reviewer's Responses to Questions

**Comments to the Authors:**

Reviewer #1: This is a nicely written prediction study, where a machine learning model used clinical and proteomics data to predict response to TNF blockers after 6 months in patients with rheumatoid arthritis. 89 patients were included. Features included demographic and clinical data and over 300 proteins that have been obtained from 60 responders and 29 non-responders. The form of the manuscript including the figures is clear.

There are some minor limitations that need to be discussed. Whilst the proteomics part is described in detail, the manuscript only provides minimal clinical information. BLDAS and gender are the most important features in the machine learning model. Why is gender so important? How about other features such as rheumatoid factor and or anti-CCP? Methotrexate is not mentioned once in the text, despite this clearly improves the response rate of TNF blockers. Was this the same in responders and non-responders? Methotrexate might also influence the protein composition in the serum. A section on limitations in the discussion (relaitvely low patient number, moderate accuracy ect.) would be appropriate.

Reviewer #2: In the present manuscript, Prasad B et al, use plasma proteomics data from a cohort of rheumatoid arthritis (RA) patients starting anti-TNF therapy. The objective pursued here is of high translational value, since there is yet no tool with which to discriminate responders from non-responder patients to this therapy. However there are several concerning aspects from the actual depth and reproducibility of the study.

- Proteomic data: One key concerning aspect is the low number of plasma samples with good proteomic data (89 from 144, 64%). The used platform, OLINK, is a robust proteomic technology and normally yields good protein quantifications. Unless the samples were extremely degraded, usable proteomic data should be obtained from a majority of the patients. Without any convincing explanation on the reason for this issue, it really questions the validity of the remaining proteomic sample data.

- Response criteria: two criteria (BSR and NICE) were used to define response. Why two methods? EULAR or ACR response criteria is a more common methods for disease response classification; what was the reason not to use these.

- Methodology: The authors describe in detail the bioinformatic packages used and have developed an accompanying R package to apply the methodology, which is useful. However, using the mentioned Github repository, there is substantial missing information describing how to apply the R package. Also, why didn’t the authors include the proteomic (and phenotype) data from the study to evaluate the method’s performance?

- Predictor: Internal cross-validation (CV) is used to develop the predictor model. This is a useful approach to estimate the generalizability of the predictor. However, the way that this has been used is highly questionable. Feature selection is done with the entire cohort of patients before CV. This includes a major bias in the procedure since it will increase the overfitting to the current dataset, despite using a cross-validation schema afterwards. Also, it is not clear how the authors balanced the unequal representation of non-responders, which is another potential bias in predictor building.

- The identification of a putative 2 class division of patients is rather arbitrary. First, the separation is done visually, with no assessment of the randomness of this separation. Second, the separating variable, PC3 is not described: how much variance is captured. Third, the fact that is associated to confounders, age and baseline DAS28, clearly does not provide much interest to the finding with regards to the objective at hand.

- No real test set is used to demonstrate the performance of the method. An application to an independent cohort of patients would really help on the credibility of the proposed patient stratification tool.

Reviewer #3: The paper by Prasad et al address an important unmet need for better patient stratification in the clinical management of arthritis. The treatments for arthritis are expensive, not universally effective and are currently administered on a trial and error basis meaning control of symptoms is sub-optimal for many patients. The study uses circulating proteins to predict response to TNFi after six months of care and identified a number of proteins that may be useful in stratification.

I have some general and specific comments.

General

It is my understanding that the raw data and the analysis methods (e.g. scripts) should be made available in an appropriate online repository. The authors state that this is achieved via a GitHub page but it is not clear to me that the raw data and analysis scripts used to perfume the analysis as presented are available in this location. If I understand correctly, this is a prerequisite for publication in PLOS Computational Biology and should be addressed.

The paper could be condensed in several sections to. For example, the introduction could be shortened and as well as the discussion (particularly between lines 91 to 104 and 455 to 472).

In the abstract the authors state “We then labelled these patients as responders (n=60) and non-responders (n=29) based on the baseline disease activity score (BLDAS).” This seems a bit misleading as EULAR criteria applied at 6-months was used?

Specific

The authors state that of 144 samples sent to Olink 1/3 (n=55) failed QC – Can the authors explain why this failure rate was so high?

The biological Importance of PCA based grouping of patients is very unclear and adds little to the paper. Could the author possibly add more detail here e.g. identify which proteins account for the perceived separation?

Could the author give more information on the TNFi therapy - was a specific TNFi used or were all TNFi permitted? Did the TNFi type differ between the responder groups? Were the patients exposed to TNFi for the first time? This information should be added to the table of patient characteristics.

My main concern is the limited sample size and the lack of true independent validation. The entire cohort included only 29 non-responders. Once you get down to the inner folds of the nested CV these numbers are starting to get extremely small. Although the authors took care to design the experiment to minimise bias due to only having a single dataset, I’m still concerned that all samples were touched by the statistical approach and no truly independent samples were available for validation. A completely separate validation sample would strengthen the paper considerably.

Due to the imbalanced nature of the outcome, did the authors consider oversampling the non-responders and if not why was this decided not to be important?

Table 1 and Figure 1. It looks like there are some quite low baseline DAS scores if I read correctly? However, I think this cannot be the case as a DAS score >5.1 at pre-treatment is an inclusion criteria?

ML methods are repeated in the results section. Please consolidate in the methods and remove from results.

**Have the authors made all data and (if applicable) computational code underlying the findings in their manuscript fully available?**

Reviewer #1: Yes

Reviewer #2: **No: **The dataset used in this study is not available. Only demo data is avaiable on the associated Github repository.

Reviewer #3: **No: **The authors say the code/data is available on GitHub. If this is the case it is not obvious to me where it is and I actually think it isn't there. This should be confirmed with the authors.

PLOS authors have the option to publish the peer review history of their article (what does this mean?). If published, this will include your full peer review and any attached files.

Reviewer #1: No

Reviewer #2: No

Reviewer #3: No
---

## [Decision Letter · Decision Letter 1]

14 May 2022

Dear Dr. Shukla,

We are pleased to inform you that your manuscript 'ATRPred: a machine learning based tool for clinical decision making of anti-TNF treatment in rheumatoid arthritis patients' has been provisionally accepted for publication in PLOS Computational Biology.

In addition, please make sure that all data and and computational code underlying the findings in the manuscript are fully available. Reviewer 1 pointed out that this is currently not the case.

Best regards,

Avner Schlessinger

Associate Editor

PLOS Computational Biology

Manja Marz

Software Editor

PLOS Computational Biology

Reviewer's Responses to Questions

**Comments to the Authors:**

Reviewer #1: All of my queries have been addressed.

Reviewer #3: I'm satisfied that the authors have addressed my comments.

**Have the authors made all data and (if applicable) computational code underlying the findings in their manuscript fully available?**

Reviewer #1: None

Reviewer #3: Yes

PLOS authors have the option to publish the peer review history of their article (what does this mean?). If published, this will include your full peer review and any attached files.

Reviewer #1: No

Reviewer #3: No

---

## [Editor Report · Acceptance letter]

24 Jun 2022

PCOMPBIOL-D-21-01368R1 

ATRPred: a machine learning based tool for clinical decision making of anti-TNF treatment in rheumatoid arthritis patients

Dear Dr Shukla,

I am pleased to inform you that your manuscript has been formally accepted for publication in PLOS Computational Biology. Your manuscript is now with our production department and you will be notified of the publication date in due course.

With kind regards,

Agnes Pap
